**Data Availability Statement:** All relevant data are within the manuscript and its Supporting Information files.

# Improved survival of non-small cell lung cancer patients after introducing patient navigation: A retrospective cohort study with propensity score weighted historic control

János G. Pitter[1], Mariann Moizs[2], Éva Somogyiné Ezer[2], Gábor Lukács[2], Annamária Szigeti[2], Imre Repa[2], Marcell Csanádi[1], Maureen P. M. H. Rutten-van Mölken[3], Kamrul Islam[4,5], Zoltán Kaló[1,6], Zoltán Vokó[1,6]*

1 Syreon Research Institute, Budapest, Hungary, 2 Moritz Kaposi General Hospital, Kaposvár, Hungary, 3 Erasmus School of Health Policy and Management, Institute for Medical Technology Assessment, Erasmus University Rotterdam, Rotterdam, The Netherlands, 4 Department of Economics, University of Bergen, Bergen, Norway, 5 NORCE-Norwegian Research Centre, Bergen, Norway, 6 Center for Health Technology Assessment, Semmelweis University, Budapest, Hungary

* voko.zoltan@semmelweis-univ.hu

## Abstract

OnkoNetwork is a patient navigation program established in the Moritz Kaposi General Hospital to improve the timeliness and completeness of cancer investigations and treatment. The H2020 SELFIE consortium selected OnkoNetwork as a promising integrated care initiative in Hungary and conducted a multicriteria decision analysis based on health, patient experience, and cost outcomes. In this paper, a more detailed analysis of clinical impacts is provided in the largest subgroup, non-small cell lung cancer (NSCLC) patients. A retrospective cohort study was conducted, enrolling new cancer suspect patients with subsequently confirmed NSCLC in two annual periods, before and after OnkoNetwork implementation (control and intervention cohorts, respectively). To control for selection bias and confounding, baseline balance was improved via propensity score weighting. Overall survival was analyzed in univariate and multivariate weighted Cox regression models and the effect was further characterized in a counterfactual analysis. Our analysis included 123 intervention and 173 control NSCLC patients from early to advanced stage, with significant between-cohort baseline differences. The propensity score-based weighting resulted in good baseline balance. A large survival benefit was observed in the intervention cohort, and intervention was an independent predictor of longer survival in a multivariate analysis when all baseline characteristics were included (HR = 0.63, p = 0.039). When post-baseline variables were included in the model, belonging to the intervention cohort was not an independent predictor of survival, but the survival benefit was explained by slightly better stage distribution and ECOG status at treatment initiation, together with trends for broader use of PET-CT and higher resectability rate. In conclusion, patient navigation is a valuable tool to improve cancer outcomes by facilitating more timely and complete cancer diagnostics. Contradictory evidence in the literature may be explained by common sources of bias, including the wait-time paradox and adjustment to intermediate outcomes.

**Funding:** The described research was conducted as part of the H2020 SELFIE project that has received funding from the European Union's Horizon 2020 research and innovation programme under grant agreement No 634288. The funders had no role in study design, data collection and analysis, decision to publish, or preparation of the manuscript. The content of this paper reflects only the SELFIE group's views and the European Commission is not liable for any use that may be made of the information contained herein. The funders had no role in study design, data collection and analysis, decision to publish, or preparation of the manuscript.

**Competing interests:** I have read the journal's policy and the authors of this manuscript have the following competing interests: the described research was conducted as part of the H2020 SELFIE project that has received funding from the European Union's Horizon 2020 research and innovation programme under grant agreement No 634288. JGP, MCs, MRM, KI, ZK, and ZV are employees of SELFIE beneficiaries. The employer of JGP, MCs, ZK, and ZV received additional EU research grants related to the evaluation of smoking cessation interventions and national cancer screening programs. MM, ÉSE, GL, and ASz are employees of Móritz Kaposi General Hospital, this Hospital initiated the OnkoNetwork program and received study funding from the SELFIE grant. These interests do not alter our adherence to PLOS ONE policies on sharing data and materials. (as detailed online in the guide for authors http://journals.plos.org/plosone/s/competing-interests).

# Introduction

## Patient navigation in cancer

Patient navigation programs aim to improve outcomes in vulnerable populations by eliminating barriers to the timely diagnosis and treatment of chronic diseases [1]. In the United States, barriers of patient access to healthcare services have strong economic components due to the lack of universal health insurance and the costs of private health insurance. The first patient navigation program in the US, launched in 1990 in the Harlem Hospital Center in New York City, was offering free or low-cost diagnostic investigations besides patient navigation to achieve timely diagnosis and treatment for breast cancer. In this flagship initiative, the proportion of patients with advanced disease (stage III-IV) at diagnosis decreased from 49% to 21%, and the 5-year survival rate increased from 39% to 70% in this hospital [1]. From that time, many other patient navigation demonstration sites have been implemented in the USA [1,2]. It's important to note that patient navigation and patient pathway management are not identical, since the latter aims at the standardization and prioritization of cancer treatment options with a focus on efficacy, safety profile, and cost-effectiveness [3]. The pathway management approach is particularly relevant in lung cancer, based on the high incidence, severity and economic costs of this cancer, and the high number of emerging new treatments with limited evidence on their cost-effectiveness [3,4].

In European countries with almost full health insurance coverage, supply limits of health systems result in waiting lists that are crucial access barriers in many settings. Many European countries which recognize the importance of timely diagnosis and treatment in cancer patients have introduced maximum care delay thresholds for cancer patients, including the UK [5] and Sweden [6]. Nevertheless, the evidence basis for introducing delay thresholds in cancer patient pathway intervals is frequently challenged in the scientific literature and the clinical consequence of care delays is not necessarily the same in different cancer types [7–10]. To better understand the apparently contradicting evidence in lung cancer studies, potential sources of methodology bias are briefly overviewed in the next section.

## Controversial findings of previous timeliness of care–overall survival studies in lung cancer

Randomized controlled studies on the clinical implications of reducing or increasing diagnostic and treatment delays in cancer care are scarce in the scientific literature, and intentional postponement of cancer treatment–that would be required in a randomized study–may be ethically disputable. Therefore, observational studies have an important role in this field. A recently published large-scale observational study investigating more than 363,000 stage I-II non-small cell lung cancer (NSCLC) patients in the US National Cancer Database found a higher 5-year mortality hazard in patients experiencing longer treatment delays (p <0.001 both in stage I and stage II groups) [11]. These robust findings concur well with previous studies reporting longer survival in early-stage lung cancer patients with shorter care delays [12–17]. However, many studies involving more advanced lung cancer cases either found no statistically significant association of care delays with patient survival [18–21], or even reported significant paradox associations: a higher mortality of patients with shorter care delays [17,22–29]. The latter, clinically counterintuitive associations are typically thought to be attributed to the 'wait time paradox' phenomenon [30], i.e., patients with the most severe/aggressive disease states receive earlier care and have worse prognoses [17,21,22,24,26,29,31].

Additional explanations for the reported paradox associations include residual confounding [31], and 'length bias' or 'immortal time bias' (patients have to survive long enough to be

diagnosed/treated) [32,33]. To define patient cohorts with more comparable baseline characteristics in future observational studies on timeliness and survival, adoption of more sophisticated methods, e.g., propensity scores or instrumental variables were recommended [31,34]. A less frequently recognized source of methodology bias is the adjustment of multivariate survival analyses to various post-baseline parameters in the putative causal chain [35], e.g., stage at diagnosis, initial treatment type, resectability, and/or resection margin status [19,24–26,36]. Importantly, faster diagnostic investigations are expected to confer survival benefits mostly via earlier diagnosis, resulting in higher resectability rates and improved stage distribution at treatment initiation. Consequently, cancer survival analyses adjusting for these intermediate outcomes are predisposed to false negative findings [37,38]. The association of diagnostic delay with overall survival is further complicated by clinical heterogeneity in the conducted investigations. Completing an additional test (e.g., PET-CT or cranial imaging) results in longer diagnostic delays [39] but may also contribute to more precise staging and treatment planning, which can be even more important than timeliness alone [31]. Hence, patient navigation programs facilitating both the timeliness and completeness of lung cancer diagnostics with standardized investigations and staging protocols [40,41] have high potentials to improve clinical prognosis.

## OnkoNetwork and the H2020 SELFIE project

OnkoNetwork is an integrated care model which offers patient navigation for adults in both inpatient and secondary outpatient care with a newly suspected solid organ cancer in the catchment area of the Moritz Kaposi General Hospital in Somogy county, Hungary. The planning, implementation and operation of OnkoNetwork have been described elsewhere [41–43]. In a nutshell, the principles of patient navigation [1] are operationalized in OnkoNetwork via 1) the development and approval of cancer-specific investigational protocols; 2) education and technical training for physician and non-physician contributors; 3) patient-level monitoring and management of timely movement from initial cancer suspect code through complete diagnostics to treatment decision by a multidisciplinary team (target: Tumor Board decision within 30 days), and then to treatment initiation (target: within 14 days after Tumor Board decision); 4) supportive IT platform that is integrated into the medical system of the hospital; and 5) nurse navigators and supervisory physicians with distinguished responsibilities. Patient navigation starts at first cancer code in the medical system, without stop criteria–however, patient navigation is most intensive until treatment initiation [41–43]. Importantly, treating physicians are free to deviate from investigation protocols, if the justification for their decision has been recorded. OnkoNetwork was initiated in 2014, and from that time it has been thoroughly evaluated in the Horizon 2020 SELFIE project (www.selfie2020.eu). The SELFIE consortium developed a taxonomy of integrated care programs for persons with multi-morbidity [44]; provided evidence-based guidelines on financing/payment schemes with adequate incentives to implement integrated care [45,46]; conducted qualitative studies for the comprehensive description of 17 promising integrated care programs in 8 EU Member States, including OnkoNetwork [41,47–51]; provided empirical evidence on the impact of integrated care for a wide range of outcomes using a Multi-Criteria Decision Analysis (MCDA) [52,53]; and developed implementation strategies tailored to different care settings and contexts in Europe, especially for Central and Eastern Europe [54]. Importantly, the SELFIE evaluation framework was designed to comply with any multi-morbidity integrated care initiative, and the MCDA evaluation criteria were formulated in generic terms to allow for the performance assessment of very different programs. Upon a closer look of the OnkoNetwork, analysis of cancer-specific clinical outcomes is warranted, by cancer type. Given that the largest subgroup of the SELFIE

OnkoNetwork study consisted of non-small cell lung cancer (NSCLC) patients, the authors first conducted a clinical evaluation in this subgroup.

## Study aims

The purpose of this study was to evaluate the impact of OnkoNetwork implementation on overall survival of NSCLC patients, and to characterize key changes in the timeliness and completeness of NSCLC care upon OnkoNetwork implementation.

## Materials and methods

### Study design and inclusion/exclusion criteria

A retrospective observational cohort study involving patients with any new solid organ cancer was conducted after the study protocol went through an ethical review and approval process by the Hungarian National Ethics Committee. (TUKEB, Decision No. 12412-2/2017/EKU). In the present NSCLC subgroup analysis, the intervention cohort was defined as all adult patients in the catchment area of the Moritz Kaposi General Hospital with new cases of NSCLC between December 2015 –November 2016. A historic control cohort was defined as all adult patients with new cases of NSCLC between September 2014 –August 2015 in the same Hospital (before the implementation of OnkoNetwork in October-November 2015). Enrollment into OnkoNetwork was not an inclusion criterion in the intervention cohort, to minimize the risk of selection bias. For both cohorts, only patients with previously undiagnosed, new NSCLC cases were enrolled, excluding patients with i) subsequently confirmed benign conditions; ii) unexpectedly short care delay (multidisciplinary tumor board meeting within 3 days and/or treatment initiation within 7 days after first cancer suspect code), since these patients probably arrived at the hospital with a suspected case of cancer that was not completely new or were very severe cases requiring immediate care; iii) patients with <30 days survival; and iv) patients lost to hospital follow-up within 30 days. Lung cancer patients with missing histology or with a mesothelioma diagnosis were also excluded (Fig 1).

### Data collection

Study data were retrospectively extracted from the prospectively maintained electronic medical system of the Hospital. Residence urbanization level (urban / rural) and residence socio-economic development score (0–100, the higher the more developed) were mapped from publicly available data by postal code regions. Clinical onset was recorded as the earliest recorded date of clinical symptoms of subsequently diagnosed lung cancer, and as the first day of the month where only the month was recorded without data on the specific day. For asymptomatic cases, clinical onset was the date of the first investigation that raised the suspect of lung cancer. Time to multidisciplinary tumor board recommendation, and time to treatment initiation (treatment delay) were calculated from the first cancer suspect code in the medical system as day 0. Cancer staging including TNM was performed based on the 7th edition of the AJCC cancer staging manual. Overall survival was measured from the date of the first lung cancer suspect code in the hospital as day 0. The end of patient follow-up period was determined as the date of death or the last appearance of the patient in the medical system of the hospital. Patients lost to follow-up were censored at their last visit. Data on the completed diagnostic investigations (bronchoscopy, chest CT, PET-CT, scintigraphy, cytology confirmation, brain imaging (MR or CT) and initiated active anticancer treatment modalities (chemotherapy, radiotherapy, surgery) were extracted both from structured data of the medical system and from free-text medical reports. Resection surgery included typical and atypical segment

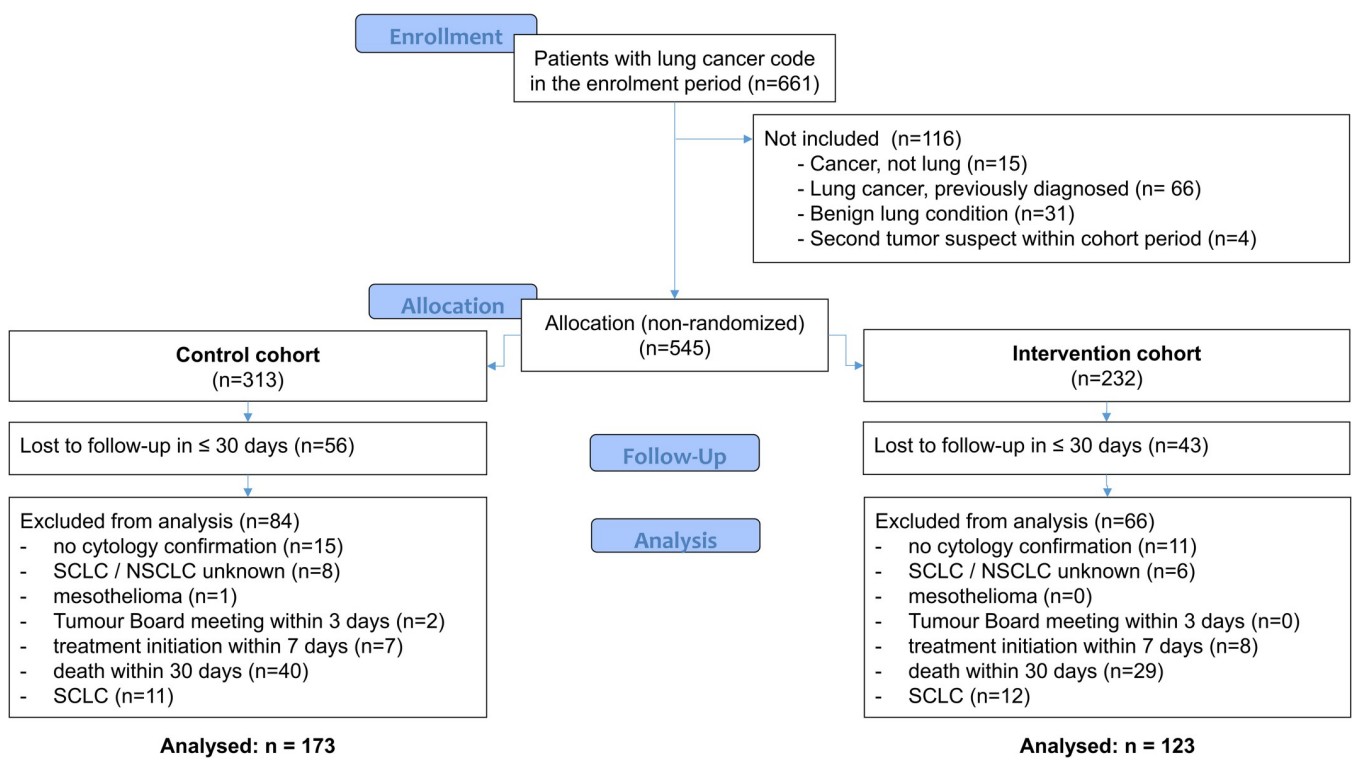

**Fig 1. Patient enrollment CONSORT flowchart.**

resection, lobectomy, and pneumonectomy, whereas metastasis resection and explorative surgery were not classified as resection surgery. Investigations completed before first suspected cancer code at the hospital (i.e., before study baseline) are referred to as "prior investigations" (e.g., prior chest CT) in manuscript text and tables. Variables on completed investigations until treatment initiation included either prior investigations or post-baseline investigations that were completed before treatment initiation. Presence or absence of investigations were evaluated as binary factors. Functional performance of patients was assessed based on routinely collected data before cancer treatment initiation using the Eastern Cooperative Oncology Group (ECOG) Performance Status scale. Largest tumor diameter measured in millimeters was extracted from pre-treatment medical records. Enrolment into OnkoNetwork was assessed based on the signed informed consent forms; enrolment was not possible for the historic control and was optional for the intervention cohort. The anonymized study data are available as supportive information (S1 File).

## Propensity score weighting

A propensity score (PS) of belonging to the intervention cohort was calculated for each patient via multivariate logistic regression with all baseline parameters included in Table 1 as predictive factors, except for prior scintigraphy which was conducted in very few patients. To estimate the average treatment effect on the treated (ATT), the inverse probability of treatment weight was set to 1 in the intervention cohort and calculated with the PS/(1-PS) formula in the control cohort, and the calculated weights were applied in all subsequent descriptive and regression analyses. This propensity score weight calculation method is also known as "standardized mortality ratio weighting" [55]. It is a well-established method of estimating

**Table 1. Baseline characteristics of the study population.** Control and intervention cohorts refer to 12-month patient samples before and after OnkoNetwork implementation, respectively. For the Intervention cohort, unweighted and weighted data are identical.

| | | Control cohort (N = 173) | | Intervention cohort (N = 123) |
| --- | --- | --- | --- | --- |
| | | Unweighted | Weighted | |
| Age mean (SD) | | 63.8 (8.9) | 64.9 (8.6) | 64.2 (8.3) |
| Sex (% male) | | 63.0% | 55.2% | 52.0% |
| Residence type (% urban) | | 51.4% | 56.1% | 56.9% |
| Residence development score (0–100), mean (SD) | | 44.2 (7.4) | 44.2 (6.9) | 44.2 (7.7) |
| Clinical onset | % asymptomatic finding | 42.8% | 39.6% | 39.0% |
| | % symptomatic | 53.2% | 58.5% | 57.7% |
| | % unknown | 4.0% | 1.9% | 3.3% |
| Tumor histology | % adenocarcinoma | 44.5% | 43.9% | 44.7% |
| | % squamous cell carcinoma | 41.6% | 43.5% | 43.1% |
| | % NSCLC neuroendocrine | 5.8% | 6.5% | 6.5% |
| | % NSCLC other/NOS | 8.1% | 6.1% | 5.7% |
| Days from onset to first hospital cancer suspect code | quartile 1 (0–13 days) | 24.9% | 19.3% | 19.5% |
| | quartile 2 (14–36 days) | 21.4% | 24.5% | 24.4% |
| | quartile 3 (37–76 days) | 19.7% | 22.8% | 24.4% |
| | quartile 4 (77–778 days) | 23.7% | 20.2% | 20.3% |
| | unknown | 10.4% | 13.2% | 11.4% |
| Completed investigations before first hospital cancer suspect code | chest CT | 68.8%* | 56.3% | 56.9% |
| | bronchoscopy | 1.7%*** | 8.6% | 11.4% |
| | PET-CT | 2.9% | 4.1% | 3.3% |
| | scintigraphy | 0.0% | 0.0% | 0.8% |
| | cytology confirmation | 0.0% | 0.0% | 0.0% |
| | brain imaging | 24.3% | 21.3% | 17.1% |
| Enrollment into OnkoNetwork | | 0%*** | 0.0%*** | 96.7% |
| Rubin's B (as compared with OnkoNetwork) | | 66.526 | 9.226 | - |
| Rubin's R (as compared with OnkoNetwork) | | 1.884 | 1.294 | - |

*p<0.05

** p<0.01

*** p< 0.001, as compared with OnkoNetwork (Chi-squared test and two-sample t-test for categorical and for normal continuous variables, respectively).

treatment effect in observational studies to ensure comparability of the treatment groups as much as possible based on observed potential prognostic covariates which might have been associated with treatment selection. Baseline balance before and after weighting was assessed by Rubin's B and R statistics as explained in the Discussion section.

## Statistical analyses

In the descriptive analyses, Chi-squared tests were conducted for categorical variables, two-sample t tests for age and largest tumor diameter, and a Kruskal-Wallis rank sum test for treatment delay. Kaplan-Meier analysis of overall survival was conducted both in the unweighted and weighted study population, with log-rank test of statistical significance. Measurable baseline characteristics (listed in Table 1), diagnostic process indicators (completed investigations, documentation of tumor stage and ECOG status, and multidisciplinary Tumor Board recommendations before treatment initiation), treatment modalities (chemotherapy, radiotherapy, resection surgery), and intermediate outcomes (TNM status, AJCC tumor stage, largest diameter, and ECOG performance status at treatment initiation) were tested as potential predictors

of overall survival in univariate and multivariate weighted Cox regression models in the weighted sample. In addition to the intervention indicator, the multivariate models included either all baseline or all post-baseline variables, in the latter model to test whether the association of intervention with overall survival was explained by improvements in intermediate outcomes. Non-significant variables were not removed from the final models. In the next step, a counterfactual analysis was performed: individual survival status at 1, 2, and 3 years was predicted from the weighted Cox regression model using intervention as the only covariate, given that the comparability of the two study groups was ensured by using propensity score weighting. For the counterfactual scenario, the survival probabilities were predicted in the intervention cohort by setting their cohort indicator parameter from "intervention" to "control" before calculating the predictions. Effect size was estimated as the mean individual difference in the predicted probabilities in the intervention versus the counterfactual cohorts. Uncertainty in the effect estimates was estimated via bootstrapping. For each outcome, 1000 bootstrapped samples were generated from the original patient sample with replacement (the same patient could be included multiple times) as if the study were repeated 1000 times. The effect estimates were calculated for these 1000 samples as they were for the original sample, and the distribution of effect size was characterized by its 95% confidence interval and by the proportion of bootstrapped samples with positive survival benefit. The counterfactual analysis, bootstrapping, and log-rank tests were conducted in Stata 16.0 [56]. Other statistical analyses were conducted in R [57] using packages survey [58], tableone [59], Desctools [60], survival [61], and survminer [62], and double-checked in Stata 16.0. All analyses were conducted on the weighted dataset, while the unweighted dataset was also analyzed in parallel with the weighted dataset at baseline balance description and in Kaplan-Meier analysis.

## Results

### Study population characteristics

Of the 661 patients with suspected lung cancer, 296 NSCLC patients were included in the analyses (123 and 173 patients in the intervention and control cohorts, respectively; Fig 1). Enrollment into OnkoNetwork was 0% and almost 100% in the control and the intervention cohort, respectively. There was no statistically significant difference across study cohorts in age, sex, residence urbanization level and socioeconomic development, tumor histology, and pre-hospital delay (time from onset to first hospital cancer suspect code), with comparable proportions of symptom-free, accidentally identified cases at clinical onset before propensity score weighting. However, the rate of completed prior chest CT and bronchoscopy investigations at study baseline showed a statistically significant difference between the cohorts (Table 1). These observed between-cohort differences were eliminated by the applied propensity score-based weighting (Table 1), resulting in Rubin's B and R values within the recommended ranges of <25 and [0.5; 2], respectively [63].

### Descriptive analyses

Post-baseline data on completed investigations and treatment procedures are summarized in S4a Table in S1 Table. The largest difference in completed investigations before treatment initiation was observed in PET-CT rates (47.2% and 37.6% in the intervention and control cohorts), followed by cytology confirmation of lung cancer (86.2% and 93.1%, respectively); however, the latter numbers do not include intraoperative cytohistology investigations. In patients not undergoing surgery, this difference was marginal (97.7% and 99.7% in the intervention and control cohorts). Surgery rate was higher in the intervention cohort (29.3% versus 23.6%), while the proportion of patients receiving chemotherapy was lower (44.7% versus

49.0%; S4a Table in S1 Table). In the two cohorts, similar proportions of patients received radiotherapy or active anticancer treatment in general. None of the above between-cohort differences reached statistical significance. The median time from first hospital cancer suspect code to treatment initiation was 9 days shorter in the intervention cohort, with a narrower interquartile range (intervention: median 58 days, IQR 36–96 days; control: median 67 days, IQR 37–112 days). Data on intermediate outcomes (tumor largest diameter, TNM stage, and ECOG status at the last tumor board meeting before treatment initiation) are shown in S4b Table in S1 Table. No significant between-cohort differences were found in these intermediate outcomes, albeit a possible trend was observed for more patients with early-stage disease in the intervention arm, as indicated by the ratio of stage I+II / stage IV cases (14.7% / 35% = 0.42 in the intervention cohort; 11.8% / 41.1% = 0.29 in the control cohort). On the other hand, the rate of patients with TNM N3 was apparently higher in the intervention cohort (25.2%, versus 15.2% in the control cohort).

## Kaplan-Meier analysis

Kaplan-Meier analysis of both the unweighted and the weighted data indicated higher overall survival in the intervention cohort, with apparent survival benefits from the first 6 months of the study (Fig 2). The difference in the unweighted analysis was not statistically significant using the log-rank test (p = 0.18), but it was statistically significant in the weighted analysis (p = 0.049).

## Cox regression analysis of overall survival

The mean follow-up duration was 1.48 years and 1.65 years in the intervention and control cohorts with 42 and 81 reported deaths, respectively. The results of weighted Cox regression

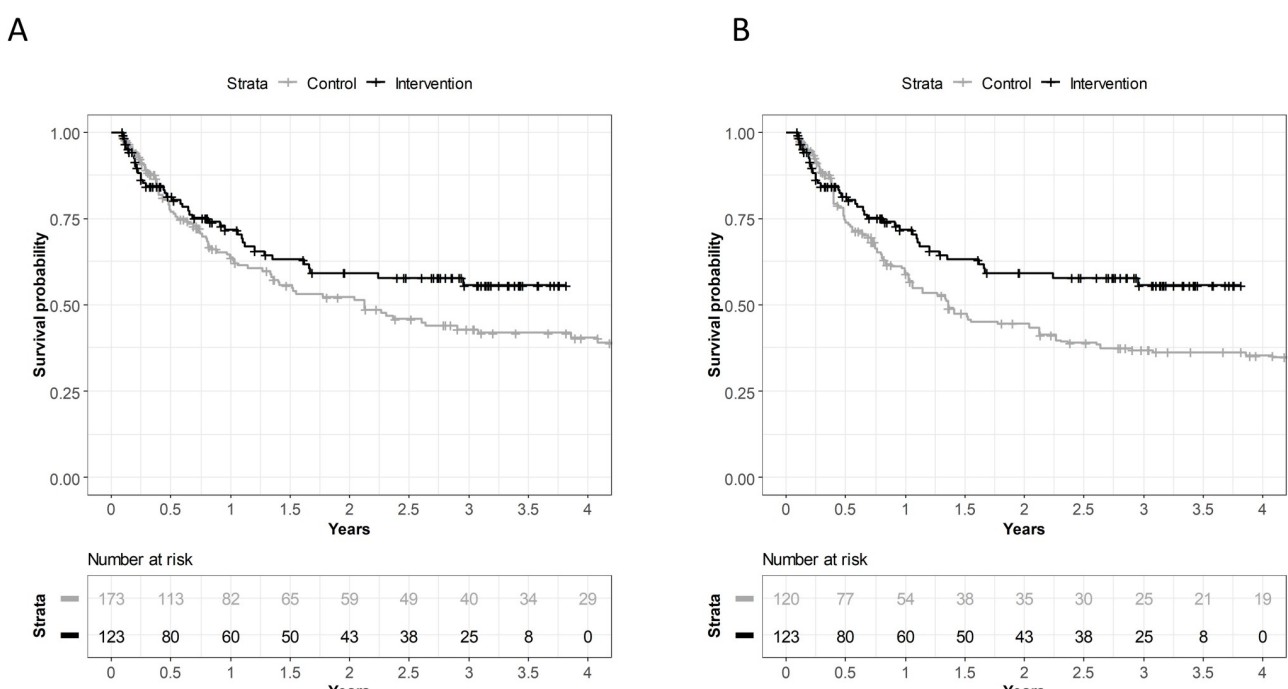

**Fig 2. Kaplan-Meier plots of overall survival of NSCLC patients.** A, unweighted data; B, weighted data. In panel B, the sample size represents the rounded sum of weights.

analyses are shown in Table 2. In univariate analyses, mortality was significantly higher in patients with a higher age, showing symptoms at clinical onset (as opposed to asymptomatic patients with accidental findings that raised the suspect of tumor), in patients with disease stage III, IV or unknown, in patients with ECOG 1, 2, or 3–4 (reference: ECOG 0), and in cases with cytology confirmation before treatment initiation. On the other hand, mortality was significantly lower in patients who belonged to the intervention cohort, who had a prior chest CT at baseline, completed a PET-CT before treatment initiation, underwent resection surgery, and who received chemotherapy. In a multivariate analysis including all baseline patient characteristics, mortality was significantly lower in patients belonging to the intervention cohort. The multivariate hazard ratio for belonging to the intervention cohort (0.63, 95%CI 0.41–0.98, p = 0.039) was practically identical to the hazard ratio found in the univariate analysis (0.64, 95%CI 0.43–0.95), indicating that the applied weighting achieved a good balance of cohorts regarding the baseline parameters. The question of whether the overall survival benefit could be explained by intermediate outcomes in the putative causal chain was further investigated in a subsequent multivariate weighted Cox regression model, adjusting for post-baseline patient characteristics (e.g., modalities of investigations conducted before treatment initiation, stage and ECOG status at treatment initiation, and treatment modalities applied, including resection surgery, chemotherapy, and radiotherapy as broad categories). As expected, belonging to the intervention cohort was not significantly associated with survival benefit in this model (HR = 0.81, p = 0.354), indicating that the included post-baseline patient path characteristics and intermediate outcomes at least partly explained the observed survival benefits of Onko-Network. Mortality was significantly lower in patients with documented AJCC TNM stage before treatment initiation (p = 0.024) and in patients receiving chemotherapy (p = 0.024), and was significantly higher in patients with stage IV disease (p<0.001) and ECOG 1 (p = 0.046) and 3–4 status (p = 0.006) at treatment initiation (Table 2). Conducting a PET-CT investigation before treatment initiation, and having a resection surgery also tended to show some association with improved survival, without statistical significance (p = 0.098 and 0.079, respectively). Descriptive patterns of the identified post-baseline predictors of overall mortality by study cohorts are depicted in Fig 3.

## Counterfactual analysis

Results of the counterfactual analysis are shown in Table 3. In these analyses, intervention cohort patients were compared to a hypothetical counterfactual cohort consisting of hypothetical patients with identical baseline characteristics, except for the assumption that counterfactual cohort patients did not receive the intervention. Overall survival was consistently higher in the intervention cohort: the intervention effect on patient survival was +12.3, +15.1, and +15.8 percent points at 1, 2, and 3 years on average, with positive survival benefits in more than 97% of the generated bootstrap samples. In other words, assuming that our study was repeated 1000 times on the population of the intervention cohort against the same hypothetical control subjects without the intervention, 977 studies would report a survival benefit in the intervention cohort at one year of follow-up. The average 1-year survival in the 1000 studies would be 71.6% in the intervention cohorts as compared to 59.3% in the control cohorts (difference 12.3%, 95%CI 0.2% - 24.9%).

## Discussion

Among all malign tumor types, lung cancer is associated with most cancer-related deaths in the world and with the highest economic costs in the European Union [64]. The highest age-standardized lung cancer incidence and mortality rates in the world were reported from

**Table 2. Association between mortality and patient characteristics (weighted dataset).** Bolded values indicate statistical significance (95%CI not including 1; note that numbers are rounded to 2 digits).

| | | Hazard ratio [95% confidence interval] | | |
|---|---|---|---|---|
| | | Univariate analyses | Multivariate model, baseline characteristics | Multivariate model, post-baseline variables |
| **Baseline characteristics** | | | | |
| Intervention cohort (reference = control cohort) | | **0.64 [0.43–0.95]*** | **0.63 [0.41–0.98]*** | 0.81 [0.52–1.26] |
| Age (years) | | **1.03 [1.00–1.06]*** | 1.02 [1.00–1.05] | - |
| Male sex (reference = female) | | 1.09 [0.72–1.63] | 0.98 [0.64–1.48] | - |
| Urban residence (reference = rural) | | 1.40 [0.92–2.14] | 1.36 [0.92–2.00] | - |
| Residence development score (scored 0 to 100) | | 1.01 [0.99–1.04] | 1.01 [0.55–1.87] | - |
| NSCLC histology$ | adenocarcinoma | 1 (reference) | | |
| | squamous cell cc | 1.07 [0.70–1.63] | 1.03 [0.66–1.63] | - |
| | neuroendocrine | 0.87 [0.38–2.00] | 0.81 [0.35–1.84] | - |
| | other / NOS | 1.10 [0.44–2.76] | 1.10 [0.48–2.55] | - |
| Clinical onset | accidental finding | 1 (reference) | | |
| | symptomatic | **1.99 [1.26–3.13]\*\*** | **1.82 [1.12–2.96]*** | - |
| | unknown | 1.98 [0.66–6.00] | 2.50 [0.77–8.12] | - |
| Time from onset to first cancer code in the hospital | 0–13 days | 1 (reference) | | |
| | 14–36 days | 0.98 [0.56–1.71] | 1.02 [0.55–1.87] | - |
| | 37–76 days | 0.94 [0.53–1.64] | 0.97 [0.53–1.77] | - |
| | 77–778 days | 0.75 [0.41–1.36] | 0.75 [0.36–1.55] | - |
| | unknown | 0.65 [0.33–1.31] | 0.67 [0.32–1.42] | - |
| Prior chest CT (reference = no) | | **0.54 [0.36–0.80]\*\*** | **0.57 [0.37–0.89]*** | - |
| Prior bronchoscopy (reference = no) | | 0.56 [0.16–1.96] | 1.19 [0.31–4.60] | - |
| Prior PET-CT (reference = no) | | 0.29 [0.05–1.62] | 0.40 [0.04–4.13] | - |
| Prior brain imaging (reference = no) | | 0.87 [0.49–1.54] | 1.39 [0.77–2.50] | - |
| **Completed investigations until treatment initiation** | | | | |
| Chest CT | | - | - | - |
| Bronchoscopy | | 1.13 [0.43–2.95] | - | 1.70 [0.70–4.05] |
| PET-CT | | **0.33 [0.22–0.49]\*\*\*** | - | 0.62 [0.36–1.09] |
| Brain imaging | | 1.12 [0.60–2.10] | - | 0.94 [0.48–1.85] |
| Cytology confirmation# | | **3.04 [1.36–6.82]\*\*** | - | 0.96 [0.32–2.92] |
| AJCC stage or TNM documented | | 1.33 [0.81–2.18] | - | **0.40 [0.18–0.89]*** |
| Tumor Board treatment recommendation | | 0.78 [0.30–2.01] | - | 0.67 [0.28–1.62] |
| **Treatment modalities** | | | | |
| Resection surgery (reference = no) | | **0.19 [0.08–0.41]\*\*\*** | - | 0.34 [0.10–1.13] |
| Chemotherapy (reference = no) | | **0.63 [0.43–0.93]*** | - | **0.59 [0.37–0.93]*** |
| Radiotherapy (reference = no) | | 1.02 [0.67–1.54] | - | 0.85 [0.52–1.38] |
| **Intermediate outcomes** | | | | |
| Stage at treatment initiation | I | Reference | | |
| | II | 1.78 [0.52–6.09] | - | 2.03 [0.59–7.02] |
| | III | **2.74 [1.09–6.88]*** | - | 1.78 [0.78–4.06] |
| | IV | **6.99 [2.98–16.39]\*\*\*** | - | **3.51 [1.69–7.29]\*\*\*** |
| | Unknown | **2.79 [1.10–7.04]*** | - | n.a. |

(*Continued*)

**Table 2.** (Continued)

| | | Hazard ratio [95% confidence interval] | | |
|---|---|---|---|---|
| | | Univariate analyses | Multivariate model, baseline characteristics | Multivariate model, post-baseline variables |
| ECOG performance at treatment initiation | 0 | Reference | | |
| | 1 | **2.44 [1.44–4.13]**\*** | - | **1.65 [1.01–2.71]**\* |
| | 2 | **3.99 [1.96–8.11]**\*** | - | 1.72 [0.71–4.17] |
| | 3–4 | **9.52 [4.88–18.57]**\*** | - | **3.62 [1.44–9.12]**\*\* |
| | unknown | 1.22 [0.32–4.71] | - | 1.36 [0.33–5.64] |

\*p<0.05

\*\* p<0.01

\*\*\* p< 0.001

$Histology is time invariant and therefore is included as baseline characteristics, albeit was unknown at baseline

#not including intraoperative evaluation.

Hungary, followed by another Central and Eastern European country, Serbia [65]. Integrated care initiatives are relatively sparse in Central and Eastern Europe in comparison to Western Europe [41]. Hence, patient navigation models with improved timeliness and quality assurance of healthcare provision are especially promising tools to improve clinical outcomes in this region. In our study, the introduction of OnkoNetwork, a patient navigation model in a

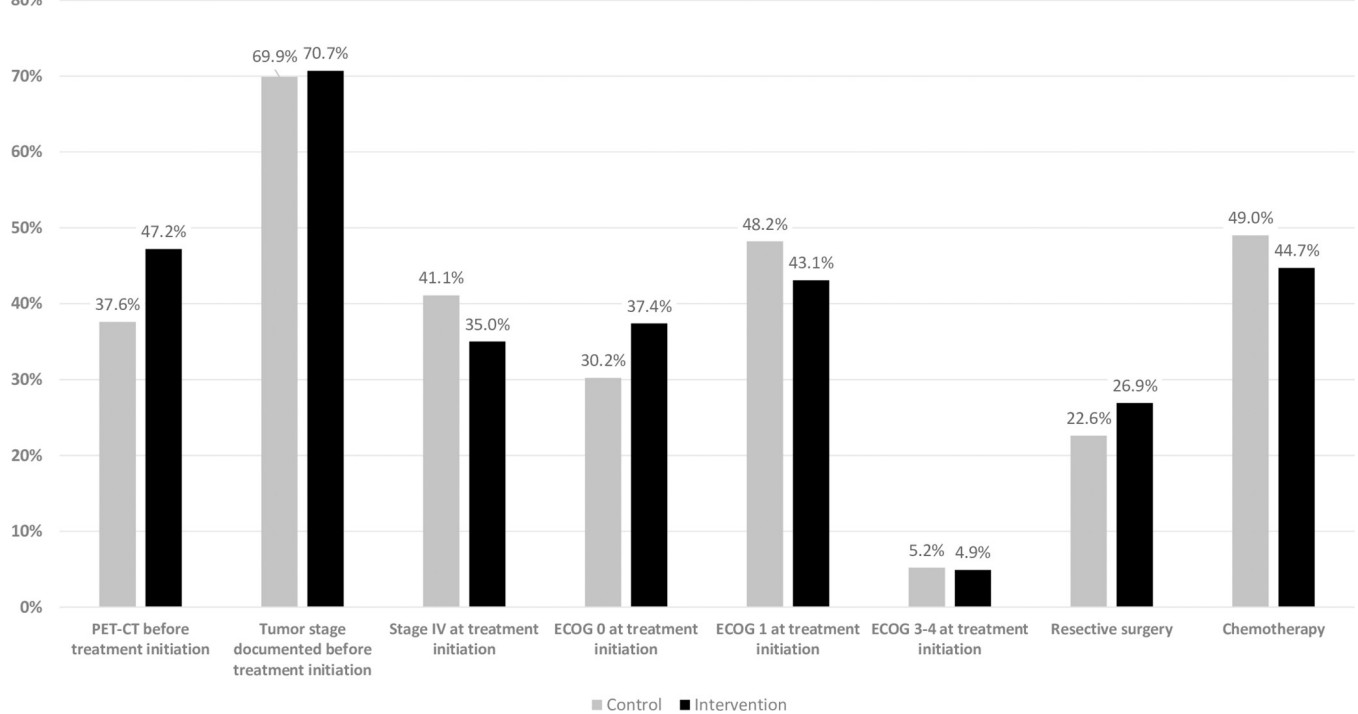

**Fig 3. Between-cohort differences in post-baseline predictors of overall survival.** The vertical axis indicates proportions of study subjects with the corresponding characteristics in the weighted dataset. For numeric data, please see S4a Table in S1 Table.

**Table 3. Counterfactual analysis of overall survival, in the hypothetical, bootstrapped weighted populations.**

| | Mean survival probability [95% CI] | | |
|---|---|---|---|
| | **1-year survival** | **2-year survival** | **3-year survival** |
| Intervention cohort | 71.6% [63.3% - 79.9%] | 59.5% [49.5% - 69.0%] | 54.2% [43.7% - 64.5%] |
| Counterfactual cohort | 59.3% [47.5% - 70.6%] | 44.4% [32.6% - 57.8%] | 38.4% [25.7% - 51.6%] |
| Effect (difference) | +12.3% [0.2%– 24.9%] | +15.1% [-0.3% - 30.4%] | +15.8% [-0.0% - 31.0%] |
| Probability of positive effect on survival | 97.7% | 97.4% | 97.5% |

Hungarian county hospital for patients with solid organ cancer, was shown to be associated with improved overall survival of lung cancer patients, preventing one of every three deaths during the 3-year follow-up period (HR 0.63, 95%CI 0.41–0.98). Interestingly, this benefit was found when comparing annual patient cohorts representing the real-world patterns of new lung cancer suspect cases in the studied county hospital, including stage I to stage IV patients.

In observational studies involving advanced lung cancer patients, various sources of selection bias may shift the results toward paradox associations of shorter care delays with worse clinical outcomes: the wait time paradox [30], length bias or immortal time bias [32,33], residual confounding [31], and adjustment of multivariate survival analyses to various post-baseline parameters in the putative causal chain [35,37,38]. Furthermore, timeliness of care must not be evaluated in isolation given that care process content can be even more important than timeliness alone [31]. As a response to these challenges, our study compared annual cohorts without patient selection based on delay durations; we adopted propensity score-based weighting to improve baseline comparability of cohorts as previously recommended [31]; and conducted a counterfactual analysis of intervention effect where the intervention cohort was compared to a hypothetical cohort consisting of the same patients who had not received the intervention.

Propensity scores are frequently applied for confounding adjustment in observational studies via matching, stratification, adjustment as a regressor, or weighting, especially in studies estimating treatment effect. An important limitation of the propensity score matching approach is that unmatched observations, even if falling within the matching caliper, are discarded from the analysis [55]. Weighting based on propensity scores has the advantages of keeping most observations in the analysis, and unlike the inclusion of propensity scores in the regression model, it allows transparent reporting of baseline balance before and after adjustment [55]. Our propensity score weighting approach ("standardized mortality ratio weighting") was selected for the target inference (average treatment effect on the treated, ATT) as harmonized across all model evaluations of the SELFIE consortium, and in compliance with current literature recommendations [55] and practice [66,67]. It is a limitation of our study that no sensitivity analyses were performed using alternative propensity score adjustment methods, but the method we followed is considered to be the best performing method among propensity score adjustment and importantly, the adopted weighting method eliminated the statistically significant baseline differences in prior chest CT and bronchoscopy rates.

Rubin's B and R values indicate the standardized mean difference and the variance ratio of propensity scores between study cohorts, respectively. Rubin suggested that B <25 and R between 0.5 and 2 indicate a sufficient overall balance [63]. In our study, baseline balance was not in the acceptable range before weighting (Rubin's B = 66.526), but a good baseline balance was achieved after propensity score weighting (Rubin's B = 9.226, Rubin's R = 1.294, and no significant difference in any of the observed baseline covariates; Table 1). Hence, the confounding effect of baseline covariates was eliminated in the weighted dataset and the effect estimates can approximate the causal effect [68]. Elimination of baseline imbalance in the

weighted dataset was also confirmed by the lack of effect of baseline parameters on the mortality estimate of intervention in the multivariate analysis in comparison to the univariate model.

Counterfactual impact evaluation is a valuable tool for comparing the outcomes of interest in subjects exposed to a program to outcomes that would have been achieved if the intervention had not been implemented [69]. In studies without randomization, the counterfactual analysis is frequently focusing on simulated outcomes that could have occurred under different conditions, e.g., without exposure to the investigated program [70,71]. In our study, the estimated hazard rate of intervention was applied in the counterfactual analysis where all baseline characteristics of the study patients showed identical patterns in the two compared cohorts, except for the assumption that counterfactual cohort patients did not receive the intervention. The predicted survival in this counterfactual, hypothetical control cohort was consistent with the observed survival in the control cohort (data in Table 3 and Fig 2, respectively). The counterfactual analysis results (Table 3) were in good agreement with the observed survival benefit associated with the introduction of OnkoNetwork, with point estimates above 10 percent point improvement at year 1, and above 15 percent point improvements in 2-year and 3-year overall survival. Furthermore, the applied bootstrapping exercise, i.e., generation of multiple hypothetical study populations via sampling from the program cohort with replacement allowed the investigation of uncertainty in the counterfactual analysis point estimates. The results presented in Table 3 indicate that the probability of positive effect on survival was above 97% at all three timepoints. This finding means that if the same study was repeated 1000 times in the same population, more than 970 of these hypothetical studies would report at least some survival benefit upon OnkoNetwork implementation. However, this analysis does not inform us about the expected statistical significance of survival benefits in those hypothetical studies.

In our study, the implementation of OnkoNetwork was associated with improved overall survival of NSCLC patients in a multivariate Cox regression model adjusted for the baseline characteristics of study participants. The multivariate hazard ratio for belonging to the intervention cohort was 0.63 (95%CI 0.41–0.98). The point estimate 0.63 indicates a large decrease in overall mortality hazard (37% decrease) which is consistent with the presented Kaplan-Meier plots in Fig 2. The statistical hypothesis testing in the weighted KM and Cox regression were consistent ($p = 0.049$ in the former and $p = 0.039$ in the latter). One cannot expect exactly the same results as these tests are not identical, the first being a nonparametric test based on the comparison of the expected and observed numbers of the outcome, the second being a Wald test of a regression coefficient.

To further investigate whether the observed association in the Cox regression model could be explained by intermediate outcomes in the putative causal chain, a subsequent multivariate weighted Cox regression model was adjusted to post-baseline patient characteristics. As expected, intervention effect was not statistically significant in the latter model, whereas mortality was significantly lower in patients with documented AJCC TNM stage before treatment initiation and in patients receiving chemotherapy, and significantly higher in patients with stage IV disease and/or ECOG 1 and 3–4 status at treatment initiation. The use of PET/CT before treatment initiation, and undergoing resection surgery also tended to be associated with longer overall survival (HR 0.62, 95%CI 0.36–1.09 and HR 0.34, 95%CI 0.10–1.13, respectively; not significant). Accordingly, the beneficial effect of OnkoNetwork implementation on lung cancer overall survival could be–at least partly–explained by a complex interplay of multiple post-baseline factors including broader exploitation of PET-CT imaging, higher proportion of patients with resection surgery, and lower proportion of advanced stage patients at treatment initiation. Indeed, some corresponding non-significant trends after OnkoNetwork implementation could be observed in descriptive analyses (Fig 3).

In our study sample, slightly fewer patients received a multidisciplinary tumor board treatment recommendations within 30 days of receiving their first hospital cancer suspect code in the intervention arm (32.5%, versus 43.1% in the control cohort, p = 0.071). However, this comparison does not take into account that the Tumor Board meetings were preceded by more detailed investigations in the intervention cohort, as evidenced by the broader utilization of PET-CT imaging before treatment initiation (47.2%, versus 37.6% in the control cohort). The majority of NSCLC patients failed to start their treatment within 44 days of their first cancer suspect code in both study cohorts (S4a Table in S1 Table). This finding may indicate that the diagnostic workup of NSCLC requires a longer time horizon than other solid organ cancers. Longer diagnostics of NSCLC than for other solid organ cancers may result from the difficulty and possible adverse effects of cytology sampling in the lung, postponement in many cases after the evaluation of nodule growth on repeated CT scans, or analysis of functional information typically via PET-CT. Nevertheless, the median time to treatment initiation was shorter in the intervention arm by 9 days (58 days, versus 67 days in the control cohort). The third quartile of treatment initiation delay was also shorter by 16 days (96 versus 112 days), indicating a less left-skewed distribution with a lower proportion of cases with extreme treatment delays in the intervention cohort. Interpretation of clinical impacts of differences in care delays is complicated by heterogeneity of the diagnostic workup procedures. Nevertheless, the general advantages of less perceived delays in oncology care are well recognized in the context of the psychological well-being of patients and patient-doctor relationship [22,23].

Our study has important limitations. First, for ethical reasons, the implementation of OnkoNetwork did not include a randomization step on patient enrollment and the adopted observational study design is subject to selection bias. Studies like these frequently face this limitation when evaluating integrated care programs, consequently, the SELFIE consortium adopted a sophisticated methodology to minimize the risk of selection bias with propensity score weighting and counterfactual analysis as core elements, which was the methodology our analysis carefully followed.

Further limitations of our study include the moderate study size, which did not allow for more robust statistical findings and more refined analyses of patient paths. The relatively broad categories of the measured patient pathway variables allowed for the definition of a manageable number of pathway events, but could hide important differences in the fine details. For more refined analyses, the use of more graded patient path indicator variables would be warranted in larger study populations, e.g., to investigate the type of chemotherapy and the number of chemotherapy episodes–instead of a high-level binary parameter whether chemotherapy was received by the patient or not. Moreover, larger study populations would allow for a lower probability of Type 1 error in statistical tests (e.g., alpha = 0.001).

Furthermore, the care delay intervals measured in our study were based on the OnkoNetwork interval definitions, with an atypical clock start (first cancer suspect code appearing in the medical system of the study Hospital). Hence, these periods cannot easily be mapped to the more and more widely used international care delay terminology established e.g. by the Aarhus statement [72]. The time from first cancer suspect code to the multidisciplinary tumor board recommendation interval in our study mostly resembles the secondary care interval minus the treatment interval as defined in the Aarhus statement.

As an additional limitation, we emphasize that our findings are linked to the Hungarian healthcare context that surrounds the OnkoNetwork model. Patient delay, length of diagnostic and treatment waitlists, and adherence of real-world clinical practice to national or institutional guidelines can be very different across countries or regions. In principle, the more room for improvement that exists in a health system in timely and quality assured care, the larger the clinical advantages that may be expected upon patient navigation system implementation, assuming that the implementation is feasible and is not blocked by legal, cultural or economic

barriers. The authors highlight the importance of qualitative and quantitative assessment of model transferability to other countries, instead of merely extrapolating the study findings to other patient populations across dissimilar healthcare settings.

Finally, our detailed analysis in lung cancer patients was limited to the evaluation of clinical outcomes. For evidence-based decision making on the implementation of new patient navigation programs, additional research is warranted on health economic aspects and on patient experience, besides confirmatory findings on improved clinical outcomes.

## Conclusions

Our study showed a promising improvement in the overall survival of NSCLC patients associated with the implementation of the OnkoNetwork patient navigation model in the Moritz Kaposi General Hospital in Kaposvár, Hungary. The positive impact of OnkoNetwork could be explained by multiple, statistically not significant beneficial trends (broader use of PET-CT during diagnostic workup, more patients diagnosed in surgically resectable stage, and slightly less advanced cases with stage IV and ECOG 3–4 status at diagnosis). Patient navigation is a valuable tool to improve lung cancer outcomes by facilitating timely and complete cancer diagnostics. An important generalizable conclusion of our study is that the selection of regressors in multivariate survival analysis should be conscious and the model results should be interpreted in accordance with the selected model structure. When including putative effect mediators as explanatory variables in the analysis of association between patient navigation and overall survival, only the remaining association independent from the included explanatory variables will be estimated. For the overall assessment of the association, regression analysis shall not be adjusted to intermediate outcomes or putative effect mediators.

## Supporting information

**S1 Table.** MS Word file, including Table S4a (Overview of diagnostic and treatment procedures, by study cohorts) and Table S4b (Intermediate outcomes at last assessment before treatment initiation, by study cohorts).
(DOCX)

**S1 File. Anonymized study data.** MS Excel file, 41 variables for the 296 included patients. Explanation for variable names and codes are included as comments.
(XLSX)

**S2 File. Statistical methods and results in R.** Statistical analysis codes and code outputs in R, in a html file.
(HTML)

**S3 File. Statistical methods and results in Stata.** MS Word file including Stata log and do files.
(DOCX)

**S4 File. STROBE checklist.** MS Word file, with page numbers and relevant text from manuscript.
(DOCX)

## Acknowledgments

The described research was conducted as part of the H2020 SELFIE project. The content of this paper reflects only the SELFIE group's views and the European Commission is not liable

for any use that may be made of the information contained herein. The Authors are very grateful to the OnkoNetwork Office staff members for their valuable contribution to data collection.

## Author Contributions

**Conceptualization:** János G. Pitter, Mariann Moizs, Imre Repa, Maureen P. M. H. Rutten-van Mölken, Kamrul Islam, Zoltán Kaló, Zoltán Vokó.

**Data curation:** Mariann Moizs, Éva Somogyiné Ezer, Gábor Lukács, Annamária Szigeti, Imre Repa.

**Formal analysis:** János G. Pitter, Marcell Csanádi, Zoltán Vokó.

**Funding acquisition:** János G. Pitter, Marcell Csanádi, Maureen P. M. H. Rutten-van Mölken, Kamrul Islam, Zoltán Kaló.

**Investigation:** János G. Pitter, Mariann Moizs, Éva Somogyiné Ezer, Gábor Lukács, Annamária Szigeti, Imre Repa, Marcell Csanádi, Maureen P. M. H. Rutten-van Mölken, Kamrul Islam, Zoltán Kaló, Zoltán Vokó.

**Methodology:** János G. Pitter, Marcell Csanádi, Maureen P. M. H. Rutten-van Mölken, Kamrul Islam, Zoltán Kaló, Zoltán Vokó.

**Project administration:** János G. Pitter, Mariann Moizs, Imre Repa, Marcell Csanádi, Maureen P. M. H. Rutten-van Mölken, Zoltán Kaló, Zoltán Vokó.

**Resources:** Mariann Moizs, Imre Repa, Marcell Csanádi, Maureen P. M. H. Rutten-van Mölken, Zoltán Kaló.

**Software:** János G. Pitter, Zoltán Vokó.

**Supervision:** Mariann Moizs, Imre Repa, Maureen P. M. H. Rutten-van Mölken, Kamrul Islam, Zoltán Kaló, Zoltán Vokó.

**Validation:** János G. Pitter, Mariann Moizs, Zoltán Vokó.

**Visualization:** János G. Pitter.

**Writing – original draft:** János G. Pitter.

**Writing – review & editing:** János G. Pitter, Mariann Moizs, Éva Somogyiné Ezer, Gábor Lukács, Annamária Szigeti, Imre Repa, Marcell Csanádi, Maureen P. M. H. Rutten-van Mölken, Kamrul Islam, Zoltán Kaló, Zoltán Vokó.

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
