## [Decision Letter · Decision Letter 0]

13 Sep 2021

PONE-D-21-01797Improved survival of non-small cell lung cancer patients after introducing patient pathway management: a retrospective cohort study with propensity score weighted historic controlPLOS ONE

Dear Dr. Vokó,

Thank you for submitting your manuscript to PLOS ONE. After careful consideration, we feel that it has merit but does not fully meet PLOS ONE’s publication criteria as it currently stands. Therefore, we invite you to submit a revised version of the manuscript that addresses the points raised during the review process.

We look forward to receiving your revised manuscript.

Kind regards,

Samer Singh, Ph.D.

Academic Editor

PLOS ONE

Additional Editor Comments (if provided):

Dear Dr Zoltan Voko,

The manuscript has been reviewed by three reviewers. Their overall comments have been positive.

However, some issues and concerns have been underlined for improving the manuscript. The comments from reviewers are appended below the message for addressal.

Additionally, elaborate on statistical (counterfactual) analysis, and why alpha = 0.05 for HR and PETCT analysis would be sufficient for decision making, given the potentially severe consequences of such a decision in the current case.

In case extra time is required to revise the manuscript and address the concerns/comments you may request it through system.

Journal Requirements:

2. In your Methods section, please ensure that all variables included in the analyses have been adequately defined.

3. Thank you for stating the following in the Competing Interests section: "I have read the journal's policy and the authors of this manuscript have the following competing interests: the described research was conducted as part of the H2020 SELFIE project that has received funding from the European Union’s Horizon 2020 research and innovation programme under grant agreement No 634288. JGP, MCs, MRM, KI, ZK, and ZV are employees of SELFIE beneficiaries. The employer of JGP, MCs, ZK, and ZV received additional EU research grants related to the evaluation of smoking cessation interventions and national cancer screening programs.  MM, ÉSE, GL, and ASz are employees of Móritz Kaposi General Hospital, this Hospital initiated the OnkoNetwork program and received study funding from the SELFIE grant."

Reviewers' comments:

Reviewer's Responses to Questions

**Comments to the Author**

1. Is the manuscript technically sound, and do the data support the conclusions?

Reviewer #1: Partly

Reviewer #2: Yes

Reviewer #3: Partly

2. Has the statistical analysis been performed appropriately and rigorously? 

Reviewer #1: I Don't Know

Reviewer #2: Yes

Reviewer #3: No

3. Have the authors made all data underlying the findings in their manuscript fully available?

Reviewer #1: Yes

Reviewer #2: Yes

Reviewer #3: No

4. Is the manuscript presented in an intelligible fashion and written in standard English?

Reviewer #1: Yes

Reviewer #2: Yes

Reviewer #3: Yes

5. Review Comments to the Author

Reviewer #1: The authors use a retrospective cohort study to evaluate the implementation of the OnkoNetwork on overall survival for patients with non-small cell lung cancer. This is a single institution study conducted at Moritz Kaposi General Hospital. The authors should be commended for rigorously evaluating a care delivery implementation that has the potential to improve care for oncology patients. I have the following recommendations for strengthening this manuscript.

1) The introduction should define in specific terms the "patient pathway management." After reading the manuscript, I was uncertain about how the authors define patient pathway management. For example, the American Society of Clinical Oncology defines a pathway as "detailed, evidence-based treatment protocols for delivering cancer care to patients with specific disease types and stages." Are the authors referring to the same type of treatment pathways? The main thrust of the introduction is around timeliness of care and biases. I think this could be shortened and further detail could be added around clinical pathways and how this study seeks to add to the pathways literature in oncology. The authors could cite various studies that have evaluated pathways in oncology and lung cancer such as DM Jackman, Journal of Oncology Practice, 2017 along with Neubauer MA, Journal of Oncology Practice, 2010.

2) The components of the OnkoNetwork are not well described. I was uncertain of what the OnkoNetwork intervention was in practice. I think the manuscript would benefit from revising lines 85 - 88 to more specifically list the components of the intervention. How is the Onkonetwork different from standard of care. Perhaps the authors could include a diagram or figure as well.

3) In lines 115 - 177, the authors write the intervention cohort was defined as patients with a "new solid tumor" diagnosis. However, wasn't it actually patients with a new non-small cell lung cancer diagnosis? I would revise to make more clear.

4) The data for the 2 cohorts are from 2014 - 2015 and 2015 - 2016. It is now 5-years old. The authors should provide some detail on why they are using data from several years ago to conduct their analysis and how this could impact their findings.

5) In the results section (line 190 - 191), the authors write that enrollment into the OnkoNetwork was 0% in the control and almost 100% in the intervention. I had assumed that to be expected given the design of the study. I was confused why the intervention group wasn't 100% enrolled in OnkoNetwork. Furthermore, do the authors need this statement given the methodology of the study design.

6) The description of the overall survival in the results could benefit from further clarification. Specifically, did the implementation of the Onkonetwork lead to the improvement in survival or did the intervention group include a greater proportion of earlier stage patients and that is what led to the improvement in survival. I would ask the authors to review lines 248 - 249 where this is discussed and add further clarity. This is also discussed in the conclusion, lines 313 - 315.

7) I think the conclusion would benefit from further discussion of how this manuscript adds to our knowledge of oncology clinical pathways and pathway management.

Reviewer #2: Overall the manuscript is well written and have addressed importance of health care management in patient outcome.

However, there are few points need to be taken care of before accepting the manuscript for publication.

This is an observational study where the authors have compared the overall survival of patients with NSCLC either have treatment delay or not. The exclusion and inclusion criteria used for the study patients are clear; however, the recruitment of patients to control and intervention cohorts were non-randomised. This process could have introduced biased in the cohort selection. The authors have used propensity score to reduce bias, yet it is important to highlight the impact of randomisation in this study.

Was the treatment regimen across the cohort uniform? If not, then it is important to discuss that how could different regiment impact on survival. In the intervention cohort, more patients underwent surgical resection, this could have contributed to the overall survival of the patients compared to those only have received chemotherapy. While comparing overall survival, uniformity in treatment regimen should be considered amongst the groups.

The quality (resolution) of the graphs need to be improved, it is difficult to read and interpret.

Reviewer #3: Overall the paper is well presented although the language should be improved to increase readability.

Major:

Line 240: The p-value of the HR is given as p=0.039. What test has been used? I assume alpha = 0.05 which then it is significant. However, in a medical context a significance level of alpha = 0.001 should be used in which case it is not significant. Discuss!

Line 250: P-value = 0.055 for PETCT. This is even not significant for alpha = 0.05 and not at all significant for alpha = 0.001.

It seems the key results of the paper show no statistical significance. Discuss!

Minor:

The language of the paper should be improved.

Section title 'Descriptive analysis'. In this section a survival analysis is discussed which is not a descriptive analysis. I suggest to change the section title.

6. PLOS authors have the option to publish the peer review history of their article (what does this mean?). If published, this will include your full peer review and any attached files.

Reviewer #1: No

Reviewer #2: No

Reviewer #3: No

---

## [Author Response · Author response to Decision Letter 0]

5 Nov 2021

The response is uploaded in a separate file.

---

## [Decision Letter · Decision Letter 1]

2 Feb 2022

PONE-D-21-01797R1Improved survival of non-small cell lung cancer patients after introducing patient navigation: a retrospective cohort study with propensity score weighted historic controlPLOS ONE

Dear Dr. Vokó,

Thank you for submitting your manuscript to PLOS ONE. After careful consideration, we feel that it has merit but does not fully meet PLOS ONE’s publication criteria as it currently stands. Therefore, we invite you to submit a revised version of the manuscript that addresses the points raised during the review process.

We look forward to receiving your revised manuscript.

Kind regards,

Samer Singh, Ph.D.

Academic Editor

PLOS ONE

Additional Editor Comments (if provided):

The study finding would be an excellent addition to the existing knowledge base. However, appropriate statistical treatment of the data remains desirable. For this reason alone, I am inclined to place it as a major revision.

The authors would like to suitably address the concerns of Reviewer number 3.

Additionally,

The statistical methods/methodology be sufficiently elaborated, and the discussion enriched accordingly to make the point.

The authors would like to be conservative while drawing the conclusion from their data.

Reviewers' comments:

Reviewer's Responses to Questions

**Comments to the Author**

1. If the authors have adequately addressed your comments raised in a previous round of review and you feel that this manuscript is now acceptable for publication, you may indicate that here to bypass the “Comments to the Author” section, enter your conflict of interest statement in the “Confidential to Editor” section, and submit your "Accept" recommendation.

Reviewer #1: All comments have been addressed

Reviewer #2: All comments have been addressed

Reviewer #3: All comments have been addressed

2. Is the manuscript technically sound, and do the data support the conclusions?

Reviewer #1: Yes

Reviewer #2: Partly

Reviewer #3: Partly

3. Has the statistical analysis been performed appropriately and rigorously? 

Reviewer #1: I Don't Know

Reviewer #2: Yes

Reviewer #3: No

4. Have the authors made all data underlying the findings in their manuscript fully available?

Reviewer #1: Yes

Reviewer #2: Yes

Reviewer #3: No

5. Is the manuscript presented in an intelligible fashion and written in standard English?

Reviewer #1: Yes

Reviewer #2: Yes

Reviewer #3: Yes

6. Review Comments to the Author

Reviewer #1: (No Response)

Reviewer #2: (No Response)

Reviewer #3: The following review relates to the statistical analysis.

Overall, the paper uses advanced methodology to address an interesting problem. The main problem of the paper is that the methodology is not motivated nor alternative approaches are discussed. Furthermore, the statistical methods are not presented in a way as needed for such an article.

Propensity score: A discussion of the propensity score needs to be presented explaining why it is needed and why there are no alternatives. Add citations to methodological papers disucussing this in detail.

Counterfactual analysis: The same needs to be provided for the counterfactual information.

Rubin's B and R statistics: Same as above.

Results section:

What are the p-values for the Kaplan Meier analysis?

For each analysis it needs to be clear what samples have been used.

A statistical hypothesis test is not a descriptive analysis! (page 10, line 191) This is wrong and needs to be corrected.

7. PLOS authors have the option to publish the peer review history of their article (what does this mean?). If published, this will include your full peer review and any attached files.

Reviewer #1: No

Reviewer #2: No

Reviewer #3: No

---

## [Author Response · Author response to Decision Letter 1]

3 Apr 2022

See in the enclosed file with the respose to reviewers

---

## [Decision Letter · Decision Letter 2]

17 Aug 2022

PONE-D-21-01797R2Improved survival of non-small cell lung cancer patients after introducing patient navigation: a retrospective cohort study with propensity score weighted historic controlPLOS ONE

Dear Dr. Vokó,

Thank you for submitting your manuscript to PLOS ONE. The reviews of the manuscript are appended below. Some minor clarification regarding the sample size, statistical analysis and its presentation as raised by the reviewers is required. Therefore, we invite you to submit a revised version of the manuscript that addresses the points raised during the review process.

We look forward to receiving your revised manuscript.

Kind regards,

Samer Singh, Ph.D.

Academic Editor

PLOS ONE

Journal Requirements:

Reviewers' comments:

Reviewer's Responses to Questions

**Comments to the Author**

1. If the authors have adequately addressed your comments raised in a previous round of review and you feel that this manuscript is now acceptable for publication, you may indicate that here to bypass the “Comments to the Author” section, enter your conflict of interest statement in the “Confidential to Editor” section, and submit your "Accept" recommendation.

Reviewer #2: (No Response)

Reviewer #4: (No Response)

2. Is the manuscript technically sound, and do the data support the conclusions?

Reviewer #2: Yes

Reviewer #4: Yes

3. Has the statistical analysis been performed appropriately and rigorously? 

Reviewer #2: Yes

Reviewer #4: Yes

4. Have the authors made all data underlying the findings in their manuscript fully available?

Reviewer #2: Yes

Reviewer #4: Yes

5. Is the manuscript presented in an intelligible fashion and written in standard English?

Reviewer #2: Yes

Reviewer #4: Yes

6. Review Comments to the Author

Reviewer #2: Minor revision:

For counterfactual analysis, please mention if the baseline characteristics were selected from control cohort or intervention cohort.

If selected from control cohort, is it possible that the survival benefit difference may increase or reduce between intervention and hypothetical cohort?if yes, then one of the possible explanation may be that owing to the propensity score matching using control group baselines characteristics may not make a difference for the analysis. Is there any other possible explanation? This should be discussed in the discussion.

Reviewer #4: The authors present results from a study on the impact of patient navigation for non-small cell lung cancer patients in Hungary on survival. Authors use propensity score weighting to address imbalance in the group that had access to patient navigation and the historical control cohort that did not have access to patient navigation. Authors demonstrate improved survival in the group that had patient navigation, though this difference did not remain after accounting for post-baseline clinical measures (which showed some potential imbalance (though not statistically significant) even after weighting). The manuscript will be strengthened if authors consider the following points.

1. To avoid confusion between Table 1 sample sizes and the at risk numbers given for the K-M curves in Figure 2 for the weighted sample, authors may want to clarify that the weighted sample size is 120.

2. It is not clear to me why authors included all of the baseline variables in the Cox model, since the propensity score weighting was done to approximately balance these variables between groups.

3. Throughout the manuscript, authors state that groups are "similar" or distributions are "similar". A non-significant p-value does not prove a lack of difference between groups. This is a minor point, since readers can refer to the tables to evaluate how "similar" they feel groups are on certain characteristics, but authors should keep this in mind when reporting results.

4. In my first read of the manuscript, I was a bit confused by Figure 3, since it wasn't clear to me what percentages were being shown. I thought initially it had to do with survival in the groups among those who had particular characteristics, since this was being shown after the Cox results. I realized that these are just the observed/weighted percentages in the two groups (as is presented in Supplementary Table S4a). Authors may want to refer readers to this table when mentioning Figure 3 in the text, to clarify.

Minor edits:

1. line 82: authors need to add a ")" after "groups" to close the parentheses from earlier in the sentence.

2. line 162: "Study data was" should be "Study data were"

3. line 190: "study data is" should be "study data are"

4. lines 238, 240: STATA should be Stata: https://www.statalist.org/forums/help#spelling

5. line 239: "using a" should just be "using"

6. Table 2: There are a couple of significant results that are not bolded, and a non-significant result that is bolded. Authors should be consistent in their use of bolded results. Also authors are missing a "]" for the CI for Urban residence in the multivariate baseline characteristics model.

7. line 397: "(Table 3) was" should be "(Table 3) were"

7. PLOS authors have the option to publish the peer review history of their article (what does this mean?). If published, this will include your full peer review and any attached files.

Reviewer #2: No

Reviewer #4: No

---

## [Author Response · Author response to Decision Letter 2]

23 Sep 2022

Attached as a separate file, but please, find its content here as well.

We thank the Editor and the Reviewers for their helpful comments on our manuscript. Below is our response to each additional point raised by Reviewer #2 and Reviewer #4. We hope that the corresponding revisions and the provided justifications are satisfactory and that the manuscript will be now suited for publication in PLOS ONE.

Our responses to Reviewers are typed in Italic, and changes in manuscript text are typed in red with citation marks below. 

Sincerely, on behalf of all co-authors,

Prof. Zoltán Vokó

Reviewer #2:

For counterfactual analysis, please mention if the baseline characteristics were selected from control cohort or intervention cohort. If selected from control cohort, is it possible that the survival benefit difference may increase or reduce between intervention and hypothetical cohort? If yes, then one of the possible explanation may be that owing to the propensity score matching using control group baselines characteristics may not make a difference for the analysis. Is there any other possible explanation? This should be discussed in the discussion.

Response: Thank you for your considerations on this interesting aspect of the presented counterfactual analyses. We agree that difference between the intervention and a hypothetical intervention cohort would be worth discussing, in case the baseline characteristics for the counterfactual analyses were selected from the control cohort. However, in the presented counterfactual analyses, the survival probabilities were predicted in the intervention cohort by setting their cohort indicator parameter from “intervention” to “control” before calculating the predictions (as already mentioned in lines 228-231 of the manuscript). Accordingly, the hypothetical cohort is a pseudo-control cohort in this study. This is why the difference between the intervention cohort and the counterfactual cohort was interpreted as an effect estimate of the investigated intervention in Table 3. 

Nevertheless, following the logic of your suggestion, comparing the counterfactual, pseudo-control cohort to the true control cohort might also be interesting for the readers. Mean survival probability in the counterfactual cohort at 1, 2, and 3 years was consistent with the Kaplan-Meier plot of the control cohort. Accordingly, we added a corresponding statement to the discussion section, as follows (typed in red):

“The predicted survival in this counterfactual, hypothetical control cohort was consistent with the observed survival in the control cohort (data in Table 3 and Figure 2, respectively).”

Reviewer #4:

The authors present results from a study on the impact of patient navigation for non-small cell lung cancer patients in Hungary on survival. Authors use propensity score weighting to address imbalance in the group that had access to patient navigation and the historical control cohort that did not have access to patient navigation. Authors demonstrate improved survival in the group that had patient navigation, though this difference did not remain after accounting for post-baseline clinical measures (which showed some potential imbalance (though not statistically significant) even after weighting). The manuscript will be strengthened if authors consider the following points.

Response: Thank you for your review, and please find below our response to the specific points raised. 

1. To avoid confusion between Table 1 sample sizes and the at risk numbers given for the K-M curves in Figure 2 for the weighted sample, authors may want to clarify that the weighted sample size is 120.

Response: Thank you for your suggestion, the legend of Figure 2 has been revised accordingly:

“Fig 2. Kaplan-Meier plots of overall survival of NSCLC patients. A, unweighted data; B, weighted data. In panel B, the sample size represents the rounded sum of weights.” 

In addition, control cohort sample size and total weight are also clarified and made explicit in the supplementary tables. Accordingly, a revised “S4 Supplementary tables” file is attached to the revised manuscript version. 

2. It is not clear to me why authors included all of the baseline variables in the Cox model, since the propensity score weighting was done to approximately balance these variables between groups.

Response: Indeed, the propensity score weighting was done to approximately balance the baseline variables between the study cohorts. Hence, it was assumed that a univariate Cox regression analysis of intervention effectiveness in the weighted dataset could be justified. A corresponding univariate analysis is reported in Table 2. 

Our main rationale to present also a multivariate analysis adjusted for all baseline variables was that from this model, hazard ratios for other baseline characteristics could also be evaluated, supporting the reader to put our findings into context regarding other independent risk factors of mortality in the NSCLC population. Furthermore, not presenting the multivariate model could be interpreted as a methodology limitation of our study and could result in exclusion of our findings from future systematic reviews and meta-analyses. Hence, the Authors felt that including both the univariate and the multivariate Cox model on baseline predictors have additional value, although we acknowledge that these models may be considered redundant when solely the intervention effectiveness is considered. 

3. Throughout the manuscript, authors state that groups are "similar" or distributions are "similar". A non-significant p-value does not prove a lack of difference between groups. This is a minor point, since readers can refer to the tables to evaluate how "similar" they feel groups are on certain characteristics, but authors should keep this in mind when reporting results.

Response: Thank you for your observation, we fully concur with your comment. The manuscript has been checked and all corresponding sentences are refined in the current revision, as follows:

“Study population characteristics

Of the 661 patients with suspected lung cancer, 296 NSCLC patients were included in the analyses (123 and 173 patients in the intervention and control cohorts, respectively; Fig 1). Enrollment into OnkoNetwork was 0% and almost 100% in the control and the intervention cohort, respectively. The study cohorts showed similar distributions by There was no statistically significant difference across study cohorts in age, sex, residence urbanization level and socioeconomic development, tumor histology, and pre-hospital delay (time from onset to first hospital cancer suspect code), with similarcomparable proportions of symptom-free, accidentally identified cases at clinical onset before propensity score weighting. However, the rate of completed prior chest CT and bronchoscopy investigations at study baseline showed a statistically significant difference between the cohorts (Table 1). These observed between-cohort differences were eliminated by the applied propensity score-based weighting (Table 1), resulting in Rubin’s B and R values within the recommended ranges of <25 and [0.5; 2], respectively [63].”

…

“In the two cohorts, similarcomparable proportions of patients received radiotherapy or active anticancer treatment in general. None of the above between-cohort differences reached statistical significance.”

…

“This finding means that if the same study was repeated 1000 times in similar the same populations, more than 970 of these hypothetical studies would report at least some survival benefit upon OnkoNetwork implementation.”

4. In my first read of the manuscript, I was a bit confused by Figure 3, since it wasn't clear to me what percentages were being shown. I thought initially it had to do with survival in the groups among those who had particular characteristics, since this was being shown after the Cox results. I realized that these are just the observed/weighted percentages in the two groups (as is presented in Supplementary Table S4a). Authors may want to refer readers to this table when mentioning Figure 3 in the text, to clarify.

Response: Thank you for your point, the axis label for Figure 3 has been supplemented to facilitate the proper interpretation of the figure:

“Fig 3. Between-cohort differences in post-baseline predictors of overall survival. The vertical axis indicates proportions of study subjects with the corresponding characteristics in the weighted dataset. For numeric data, please see Table S4a. “

Cross-checking the numeric data in supplementary tables with the revised statistical output file submitted in March 2022, a few typos were identified in the supplementary tables and corrected as well. Specifically: the control sample size was mistyped in Table S4b; proportions receiving any oncology treatment within 44 days after first cancer code (Table S4a), and largest tumor diameter at last assessment before treatment initiation (Table S4b) have also been corrected. These minor typo corrections did not affect the observed trends, neither the significance of between-group comparisons. The corrected supplementary tables file is part of the revised manuscript package. 

Minor edits:

1. line 82: authors need to add a ")" after "groups" to close the parentheses from earlier in the sentence.

2. line 162: "Study data was" should be "Study data were"

3. line 190: "study data is" should be "study data are"

4. lines 238, 240: STATA should be Stata: https://www.statalist.org/forums/help#spelling

5. line 239: "using a" should just be "using"

6. Table 2: There are a couple of significant results that are not bolded, and a non-significant result that is bolded. Authors should be consistent in their use of bolded results. Also authors are missing a "]" for the CI for Urban residence in the multivariate baseline characteristics model.

7. line 397: "(Table 3) was" should be "(Table 3) were"

Response: Thank you for the careful review of our manuscript, the corresponding changes are implemented in the manuscript files. 

Regarding bolded results in Table 2, we made the proposed corrections, with a single exception which requires additional justification: The 95% confidence interval of HR for age in the multivariate baseline model included 1, therefore, it was not considered significant and is not typed in bold. This is apparently in conflict with the significance of age in the univariate analysis, due to rounding of confidence interval limits to 2 digits in the table. Accordingly, the table legend has also been supplemented as follows:

“Table 2. Association between mortality and patient characteristics (weighted dataset). Bolded values indicate statistical significance (95%CI not including 1; note that numbers are rounded to 2 digits).” 

We thank again for all the constructive and helpful comments of the Editor and the Reviewers to increase the scientific merit of our manuscript, and strongly hope that the revised manuscript will meet the high standards of PLOS ONE for publication.

---

## [Decision Letter · Decision Letter 3]

13 Oct 2022

Improved survival of non-small cell lung cancer patients after introducing patient navigation: a retrospective cohort study with propensity score weighted historic control

PONE-D-21-01797R3

Dear Dr. Vokó,

We’re pleased to inform you that your manuscript has been judged scientifically suitable for publication and will be formally accepted for publication once it meets all outstanding technical requirements.

Kind regards,

Samer Singh, Ph.D.

Academic Editor

PLOS ONE

Additional Editor Comments (optional):

Reviewers' comments:

Reviewer's Responses to Questions

**Comments to the Author**

1. If the authors have adequately addressed your comments raised in a previous round of review and you feel that this manuscript is now acceptable for publication, you may indicate that here to bypass the “Comments to the Author” section, enter your conflict of interest statement in the “Confidential to Editor” section, and submit your "Accept" recommendation.

Reviewer #4: All comments have been addressed

2. Is the manuscript technically sound, and do the data support the conclusions?

Reviewer #4: (No Response)

3. Has the statistical analysis been performed appropriately and rigorously? 

Reviewer #4: (No Response)

4. Have the authors made all data underlying the findings in their manuscript fully available?

Reviewer #4: (No Response)

5. Is the manuscript presented in an intelligible fashion and written in standard English?

Reviewer #4: (No Response)

6. Review Comments to the Author

Reviewer #4: (No Response)

7. PLOS authors have the option to publish the peer review history of their article (what does this mean?). If published, this will include your full peer review and any attached files.

Reviewer #4: No

---

## [Editor Report · Acceptance letter]

17 Oct 2022

PONE-D-21-01797R3 

Improved survival of non-small cell lung cancer patients after introducing patient navigation: a retrospective cohort study with propensity score weighted historic control 

Dear Dr. Vokó:

I'm pleased to inform you that your manuscript has been deemed suitable for publication in PLOS ONE. Congratulations! Your manuscript is now with our production department. 

Kind regards, 

on behalf of

Dr Samer Singh 

Academic Editor

PLOS ONE